



# Tracking Marine Heatwaves in the Balearic Sea: Temperature Trends and the Role of Detection Methods

Blanca Fernández-Álvarez, Bàrbara Barceló-Llull, Ananda Pascual

Institut Mediterrani d'Estudis Avançats, IMEDEA (CSIC-UIB), Esporles, Spain

5 *Correspondence to:* Blanca Fernández-Álvarez (bfernandez@imedea.uib-csic.es)

**Abstract.** Marine heatwaves (MHWs) are defined as discrete periods of anomalous ocean warming. In the most commonly used MHW determination method, the threshold over which a certain temperature is considered a MHW is calculated using a fixed baseline constructed from a common climatology (1982-2001). By this definition, these phenomena have been increasing in frequency and intensity due to global warming, and it is expected to ultimately lead to a saturation point. Significant efforts 10 have been directed towards developing new ways of defining marine heatwaves motivated by the need to differentiate between long-term temperature trends and extreme events. The Mediterranean Sea serves as an ideal backdrop for comparing different MHW detection methods due to its rapid response to climate change, with higher warming trends than the global ocean. In this work, we evaluate sea surface temperature trends in the Balearic Sea, a subregion of the western Mediterranean, and compare the fixed baseline MHW detection method with two recently developed alternative methodologies. The first 15 alternative employs a moving climatology to adjust the baseline, while the second method involves detrending the temperature data before detecting MHWs with a fixed baseline. Our analysis reveals a statistically significant warming trend of 0.036 ± 0.001°C per year, which represents an increase of ~10% compared to previous studies in the same region due to the inclusion of two particularly warm recent years, 2022 and 2023. Regarding MHWs, all three methods identify major events in 2003 and 2022. However, the fixed baseline method indicates an increase in MHW frequency and duration over time, a tendency not 20 detected by the other methodologies, since we are isolating the extreme events from the long-term warming trend. This study underscores the importance of selecting an appropriate MHW detection method that aligns with the intended impact assessments. Studies performed with a moving baseline or detrended data could be more appropriate to analyse species with higher adaptability, while a fixed baseline could be a better option to study species less adaptable and more sensitive to exceeding a critical temperature threshold.

## 1 Introduction

Oceans play a critical role in regulating Earth's climate through the transport of heat and carbon across the globe. It is calculated that the ocean uptakes over 90% of the anthropogenic excess heat (Glecker et al., 2016) and around 30% of the anthropogenic carbon (Gruber et al., 2019) produced since the Industrial Revolution. This comes accompanied by an increase in the overall ocean temperature, which, in turn, can modify the sea level and ocean circulation (IPCC, 2022). Understanding how the oceans



respond to global change and how they impact the climate system is one of the biggest challenges scientists face nowadays (Pascual and Macías, 2021).

One consequence of ocean warming is the increase in frequency and intensity of marine heatwaves (MHWs) (Oliver, 2019). The study of MHWs is relatively new and has evolved into a rapidly growing field of research (for reviews, see Oliver et al., 2021; Capotondi et al., 2024). The term was introduced by Pearce et al. in 2011, and the first widely accepted definition was

proposed by Hobday et al. in 2016. MHWs are defined as discrete periods of anomalous warming of the sea. In the past half-decade, significant efforts have been devoted to characterising their main drivers, a challenging task due to their complexity and diversity. The processes influencing their formation are interconnected and operate across a broad range of temporal and spatial scales, from local dynamics affecting the mixed layer temperature budget to large-scale climate modes that enhance their frequency and persistence (Holbrook et al., 2019).

The increase in MHW frequency over time has been demonstrated in studies in studies (e.g. Oliver et al. 2018) reporting an increment of up to 54% in MHW days per year from 1925 to 2016. Projections of future ocean temperatures indicate that this increase will continue under global warming scenarios (Oliver, 2019). Moreover, recent research by Oliver et al. (2021) suggests that MHWs may reach a saturation point in the near future.

All the studies previously mentioned detect MHWs using the most commonly applied method, based on a fixed baseline

constructed from a common climatology (1982-2001) to compute the threshold over which a certain temperature is considered a MHW (Hobday et al., 2016). This method is sensitive to ocean warming because it is based on a historical climatology; as a result, events of recent years appear more magnified due to the underlaying ocean warming (Oliver, 2019). To overcome this limitation, efforts have been made in recent years to develop new methods for defining MHWs that distinguish between long-term temperature trends and extreme events (Capotondi et al., 2024). Two approaches have been proposed as alternatives to

the fixed baseline method. The first alternative uses a moving baseline, meaning that the climatology is updated annually (Rosselló et al., 2023). With this approach, the baseline shifts to only include data from the 20 years prior to the year under study. This ensures that the climatology reflects conditions characteristic of that specific period, allowing the identification of MHWs relative to recent climate conditions. The second approach involves detrending the SST data prior to applying the MHW detection process (Martínez et al., 2023). By removing the long-term warming trend, this method ensures that MHWs

are identified independently of progressive ocean warming.





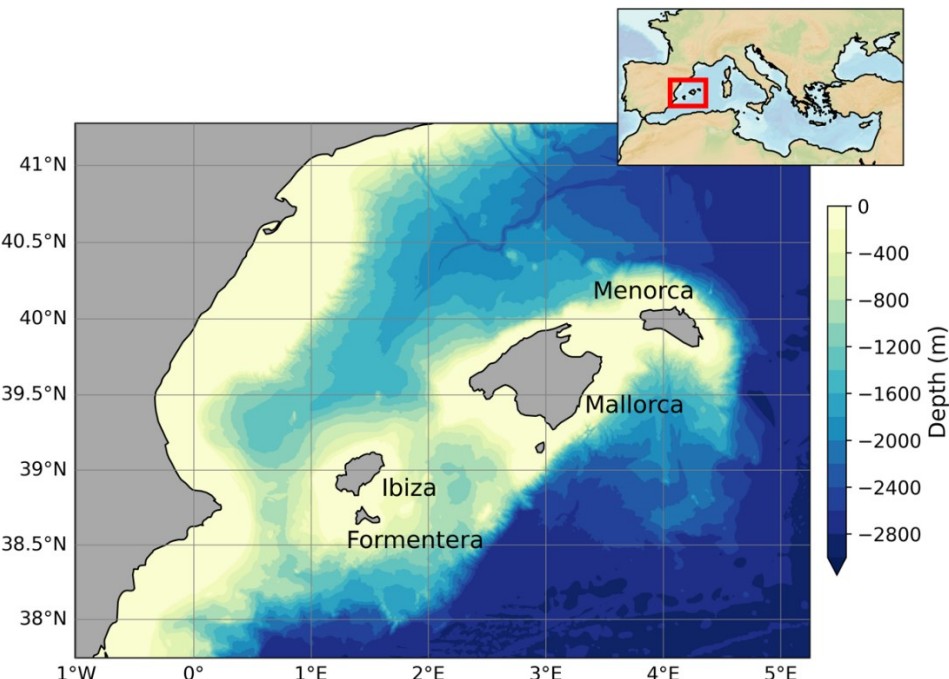

**Figure 1: Bathymetry of the Balearic Sea. Its position relative to the Mediterranean Sea is indicated with a red box in the inset. The topography was obtained from the GEBCO 15 arc-second 2024 Grid (doi: 10.5285/1c44ce99-0a0d-5f4f-e063-7086abc0ea0f).**

The Mediterranean Sea is considered a region with a rapid response to climate change, with warming trends ~20% higher than in the global ocean (Lionello and Scarascia, 2018; Juza and Tintoré, 2021). Under a warming ocean scenario and using the fixed baseline MHW detection method (Hobday et al., 2016), it is expected that the frequency and intensity of MHWs will undergo a progressive increase (Rosselló et al., 2023). This pattern has already been observed in the Mediterranean Sea, as noted by Juza et al. (2022), showing an increase of 78.4 MHW days per year from 1982 to 2020. MHWs in this region are mainly generated due to long periods of anomalously high atmospheric pressure that lead to weakened values of wind speed (Darmaraki et al., 2024). The wind reduction can alter the local air-sea heat fluxes and producing a decrease in ocean heat loss and result in the formation of MHWs. Furthermore, these events can be intensified or modified by local oceanic and weather processes (Darmaraki et al., 2024). In the Mediterranean Sea, MHWs can impact the growth, survival, fertility, migration, and phenology of both pelagic and benthic organisms, even leading to mass mortality events (Marbà et al., 2015).

This study focuses on the Balearic Sea, a subregion of the western Mediterranean that surrounds the Balearic Islands (Fig. 1). This area is one of the ecoregions with the greatest risk of species loss due to climate change (Chatzimentor et al., 2023) and processes occurring around the Balearic Islands (Aguiar et al., 2022) can impact commercially important fish stocks (Heslop et al., 2012), including the Atlantic bluefin tuna (Alemany et al., 2010; Reglero et al., 2018). The objective of this study is to evaluate Sea Surface Temperature (SST) trends and MHWs detected using different methods in the Balearic Sea through the



analysis of 42 years (1982-2023) of SST satellite observations. To the best of our knowledge a quantitative comparison of the
selected MHW detection methods has never been done before.

## 2 Data and methods

### 2.1 Sea surface temperature observations

The SST satellite-based observations analysed in this study are obtained from a level 4 (L4) product, and consequently, produced by the optimal interpolation of different sensors' observations. We use a product distributed by the Copernicus Marine Monitoring Service (CMEMS) (Le Traon et al., 2019), with the identifier SST_MED_SST_L4_REP_OBSERVATIONS_010_021 (DOI: 10.48670/moi-00173). This SST product is a daily nighttime dataset consisting of a regular grid with a resolution of 0.05° over the Mediterranean Sea (Merchant et al., 2019). The temporal extent of this product spans from 25 August 1981 to one month before the present day. For the purpose of this study, the data used ranges from 1 January 1982 to 31 December 2023, obtaining, therefore, 42 complete years of SST data.

### 2.2 Computation of SST trends

The SST trends were determined with the Theil-Sen slope estimator (Helsel et al., 2020) after removing the seasonal cycle from the original SST data. Their statistical significance (p-value < 0.05) was assessed using the modified Mann–Kendall test, which accounts for autocorrelation within the time series (Yue and Wang, 2004). The standard errors of the SST trends were calculated as the residual standard error divided by the square root of the sum of squared differences in the independent variable (James et al., 2023).

### 2.3 Detection of marine heatwaves

#### 2.3.1 Fixed baseline method

Following the definition proposed by Hobday et al. (2016), a MHW is described as a high-temperature event where a threshold value is exceeded for a minimum of five consecutive days. The threshold is usually set at the 90th percentile of the distribution at each grid point over a specific climatology. The fixed baseline refers to the use of a common climatology in the MHW identification process, computed here using data from the period between 1982 and 2001 (Hobday et al., 2016; Darmaraki et al., 2024). Considering the fixed baseline is the most widespread definition, we will use it as a benchmark to compare with other methodologies.

#### 2.3.2 Alternative methods: moving baseline and detrending

Two alternatives for detecting MHWs are the moving baseline method (Rosselló et al., 2023) and the detrending approach (Martínez et al., 2023). In the moving baseline method, the climatology is updated annually, using data from the 20 years



preceding the year under study (Rosselló et al., 2023). For instance, the climatology used to detect MHWs in 2002 is computed with data from 1982 to 2001, while to detect MHWs in 2023, the climatology is computed with data from 2003 to 2022. Following this approach, the threshold is adapted to account for evolving climate conditions. To account for the initial 20 years

required to build the climatology, we begin detecting MHWs in 2002 and continue through 2023. The 20-year reference period was chosen to enable a longer analysis period (Rosselló et al., 2023), and, for consistency, is also used as the length of the reference periods in the other methods. The second approach consists of detrending the SST data prior to applying the MHW detection process (Martínez et al., 2023). The detrending is performed by subtracting the trend computed at each grid point to the original SST data. After the detrending, the MHWs are detected using a fixed baseline (1982-2001).

## 110  2.4 Marine heatwaves metrics

For each MHW event, we compute its duration and its mean, maximum and cumulative intensity (Hobday et al., 2016). The mean intensity is defined as the average SST anomaly of a MHW event, the maximum intensity is the highest SST anomaly reached during a MHW episode, and the cumulative intensity is the sum of the daily SST anomalies during a MHW event (Table 1). We also compute the total number of days per year with MHWs. These metrics are computed on each grid point.

When computing spatial averages we use a weighted mean, which allows us to consider the varying areas of the grid cells that change with latitude.

MHWs are classified into four categories according to the extent to which they exceed the 90th percentile threshold (Hobday et al., 2018):

- Category I: Moderate heatwaves (1-2 times the threshold)
- Category II: Strong heatwaves (2-3 times the threshold)
- Category III: Severe heatwaves (3-4 times the threshold)
- Category IV: Extreme heatwaves (>4 times the threshold)

**Table 1: Metrics used to characterise a MWH.**

| Parameter | | Definition | Units |
|---|---|---|---|
| MHW days | | Total number of days in a year under MHWs | [days] |
| Duration | | Number of days of a MHW event | [days] |
| Intensity | maximum | The highest SST anomaly reached during a MHW event | [°C] |
| | mean | Mean SST anomaly reached during a MHW event | [°C] |
| | cumulative | Sum of the daily SST anomalies during a MHW event | [°C] |



## 3 Sea surface temperature trend

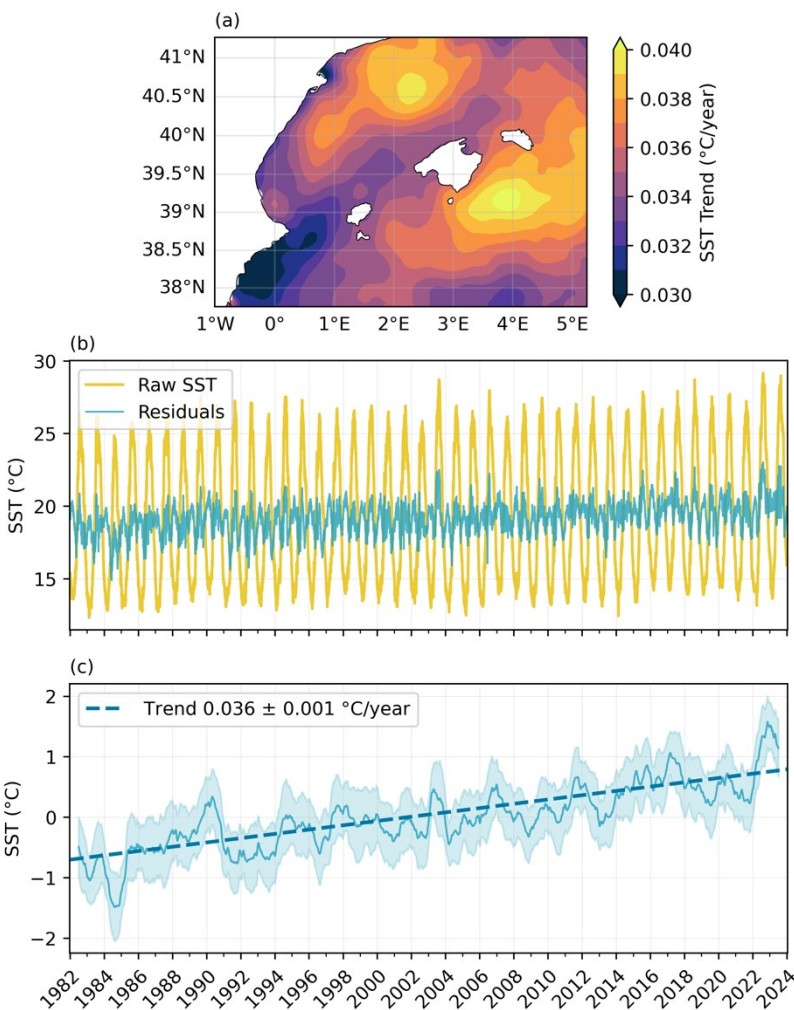

**Figure 2: (a) SST trends computed from 1982 to 2023 in the Balearic Sea. (b) Spatially averaged time series of SST (yellow) and SST residuals from the harmonic analysis (blue) with the mean value of the seasonal cycle added for readability. (c) Spatially averaged SST residuals from the harmonic analysis. For clarity, a 365-day running mean is applied (thin solid line), and the shaded areas represent the standard deviation of the mean. The SST trend is shown by a thick dashed line (trend ± standard error).**

Sea surface temperature trends computed in the Balearic Sea reveal a generalised warming between 1982 and 2023 (Fig. 2). These trends have spatial variability with minimum values of 0.030 ± 0.001°C/year located southwest of Ibiza, and maximum values of over 0.039 ± 0.001°C/year in two warming hotspots (Fig 2a). The first hotspot is located northwest of Mallorca, centred at 40.6°N 2.25°E, and the other one is located south of Menorca, centred at 39.2°N 4°E. The hotspot south of Menorca presents the maximum trends, reaching 0.041 ± 0.002°C/year. The spatially averaged SST trend for the Balearic Sea is 0.036 ± 0.001°C/year (Fig 2c).



The reported SST trends were calculated at each grid point after removing the seasonal cycle through a harmonic analysis (Sect. 2.2, Fig. 2). To compare the values obtained with this method to those derived from other methods described in the literature, we have performed a sensitivity analysis of the trend. When computing trends with a linear regression and considering complete years, the results indicate that using either the annual mean or a 365-day running average, the spatially

averaged linear regression slope is 0.036–0.037°C/year. However, when using the original (raw) data, the trend increases to 0.042°C/year. This discrepancy is due to the high sensitivity of linear regression to temporal extremes in the data, such as minimum and maximum values. If instead of using complete years the slope is calculated from the original SST data but starting and ending the time series at its yearly mean (May 1983 to May 2023), the resulting slope is 0.030°C/year—a 30% difference. We opted for using the Theil-Sen (TS) estimator on the residuals to compute the slope. Using the TS estimator

allows for a larger sample size compared to annual mean values, while accounting for data autocorrelation by incorporating the effective sample size from the modified Mann–Kendall test (Yue and Wang, 2004). This approach provides more realistic statistical significance and standard error than the 365 running averages, while producing trends consistent with both annual mean values and 365-day averages.

Previous studies in the broader western Mediterranean basin have reported SST trends consistent with those obtained in this

study for the Balearic Sea. From 1982 to 2012, Shaltout and Omstedt (2014) reported a trend of $0.035 \pm 0.007$°C/year. More recent studies by Pisano et al. (2020) and Pastor et al. (2020) reported trends of $0.036 \pm 0.006$°C/year for 1982-2018 and 0.035°C/year for 1982-2019, respectively. In the Balearic Sea, Juza and Tintoré (2021) identified a warming trend of $0.033 \pm 0.003$°C/year for 1982–2020, which is a 0.003°C/year lower than the trend computed in this study (representing an increase of ~10%). The higher trends detected in our analysis can be attributed to the inclusion of data from two particularly warm years,

2022 and 2023, during which SST and ocean heat content reached record levels in the Mediterranean Sea (Cheng et al., 2024). Including SST data from 2020 to 2023 in the analysis increases the trend from $0.033 \pm 0.001$°C/year to $0.036 \pm 0.001$°C/year.

## 4 Marine heatwaves: the effect of different detection methods

### 4.1 MHW days and mean duration

When using a fixed baseline to detect MHWs, the total number of days per year during which the Balearic Sea undergoes

MHWs increases over time (Fig. 3a). The annual number of MHW days is maximum in 2022 and 2023. In 2022, over half of the year (208 days) was under a MHW, with more than a third of these days (36%) classified as strong (category II; Fig. S1 in the Supplement). With a moving baseline, MHW days in the first part of the period (up to 2009) are similar to those obtained using a fixed baseline (Fig. 3a). However, in the latter half, the values do not increase as they do with the fixed baseline. There is no discernible tendency in the number of days experiencing MHWs or in their mean duration. Years with a high number of

days under MHWs include 2003, 2011, 2016, 2022 and 2023, with the highest being 2022, when 39% of the year was under a MHW. Among these years, the highest proportion of strong (category II) MHWs occurred in 2003 (Fig. S1 in the Supplement). Using the detrended method, the number of MHW days is the lowest across all the years analysed (Fig. 3a). The years with



the highest number of MHWs days detected with this method are 2003, 2022 and 2023, with 72.37 days, 108.25 days and 75.74 days, respectively. The largest proportion of strong MHWs is also detected in 2003 (Fig. S1 in the Supplement). A comparison of the MHW days detected with the three methods highlights differences in certain years. In 2016, both the fixed and moving baseline approaches show a high number of MHW days compared to neighbouring years, whereas the detrended method shows a dip in MHWs days. Similarly, in 2011, the detrended method detects a high number of MHWs, but this is relatively low compared to the importance this year has with the other methods.

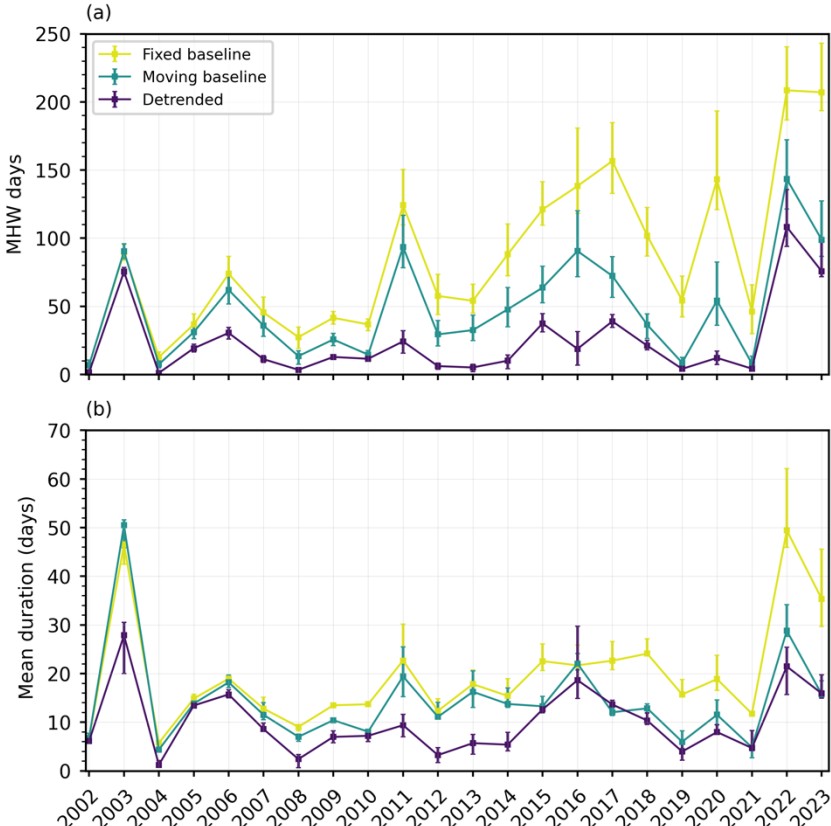

**Figure 3: Spatially averaged metrics of the MHWs detected in the Balearic Sea from 2002 to 2023 using three detection methods: fixed baseline, moving baseline and detrended data. (a) Annual number of days under MHWs. (b) Annual mean duration of the MHW events.**

The annual mean duration of MHWs also experiences an increase over time when using a fixed baseline (Fig. 3b). Two large peaks in duration are detected in 2003 and 2022, continuing, with a slight decrease, in 2023, with mean durations of 46.57, 49.45 and 35.36 days, respectively. Using a moving baseline, the mean duration in 2022 was 28.79 days, while in 2003 MHWs had an average duration of 50.47 days. With the detrended method, the highest mean duration appears in 2003 with values of





27.88 days, followed by 2022 with 21.47 days. Consistent with the results for MHW days, the detrended approach generally shows the lowest values for MHW duration when compared to the fixed and moving baselines.

## 4.2 Intensity of marine heatwaves

The annual averages of mean, maximum and cumulative intensity of MHWs averaged over the study region are shown in Fig. 5 (and Table S2 in the Supplement). With the fixed baseline, the highest mean intensity appears in 2003 with an average of 2.62°C, followed by the values obtained in 2022, 2.55°C, and 2023, 2.32°C (Fig. 4a). Despite this, the maximum intensity in 2003 (4.44°C) is exceeded in 2017 and 2022 (4.69°C and 4.76°C) and equalled in 2023 (Fig. 4b). The cumulative intensity in 2003 is exceeded in the previously mentioned years and in 2015 and 2020 (Fig. 4c). This can be attributed to the higher mean intensity of the MHWs that occurred in 2003 compared to other years. The mean intensities of MHWs in 2017 and 2022 were not as high as in 2003; however, these MHWs exhibited larger temperature anomaly peaks, resulting in these years having the highest maximum intensities. With respect to cumulative intensity, it depends not only on the temperature anomalies associated with each MHW, but also on the duration of each event. In 2015 2017, 2020, 2022 and 2023 the mean duration of MHWs was longer than in 2003 (Fig. 3b). As a result, these years had higher cumulative intensities than 2003 (Fig. 4c), even though their mean intensities were lower (Fig. 4a).

Regarding the MHWs detected with a moving baseline, only 2003, 2005, and 2022 exhibit mean intensities over 2.00°C, with values of 2.62°C, 2.06°C, and 2.15°C, respectively (Fig. 4a). Notably, with this method, the maximum intensity detected in 2003 (4.46°C) exceeds all other years by at least 0.80°C (Fig. 4b). Despite this, the cumulative intensity for 2022 surpasses that of any other year, reaching a value of 306.83°C (Fig. 4c). This indicates that while the MHWs in 2003 had the largest mean and maximum intensities, the higher mean duration of MHWs in 2022 (Fig. 3) significantly contributes to its cumulative intensity, even though its mean and maximum intensities are lower than those in 2003.

With the detrending method, 2003 and 2017 are identified as the years with the highest mean intensities, with values of 2.44°C and 2.35°C, respectively (Fig. 4a). These years are followed by 2022 with a mean intensity of 2.15°C. The years with the highest maximum intensity are also 2003 and 2017. These are the only years with the maximum intensity over 4.00°C, reaching 4.19°C and 4.04°C. Other relevant years include 2022, with maximum intensities of 4.00°C. The highest cumulative intensity is detected in 2022 (231.10°C), followed by 2003 (180.49°C) and 2023 (150.89°C) (Fig. 4c). These three years showed cumulative intensities at least 55°C higher than the rest of the years.





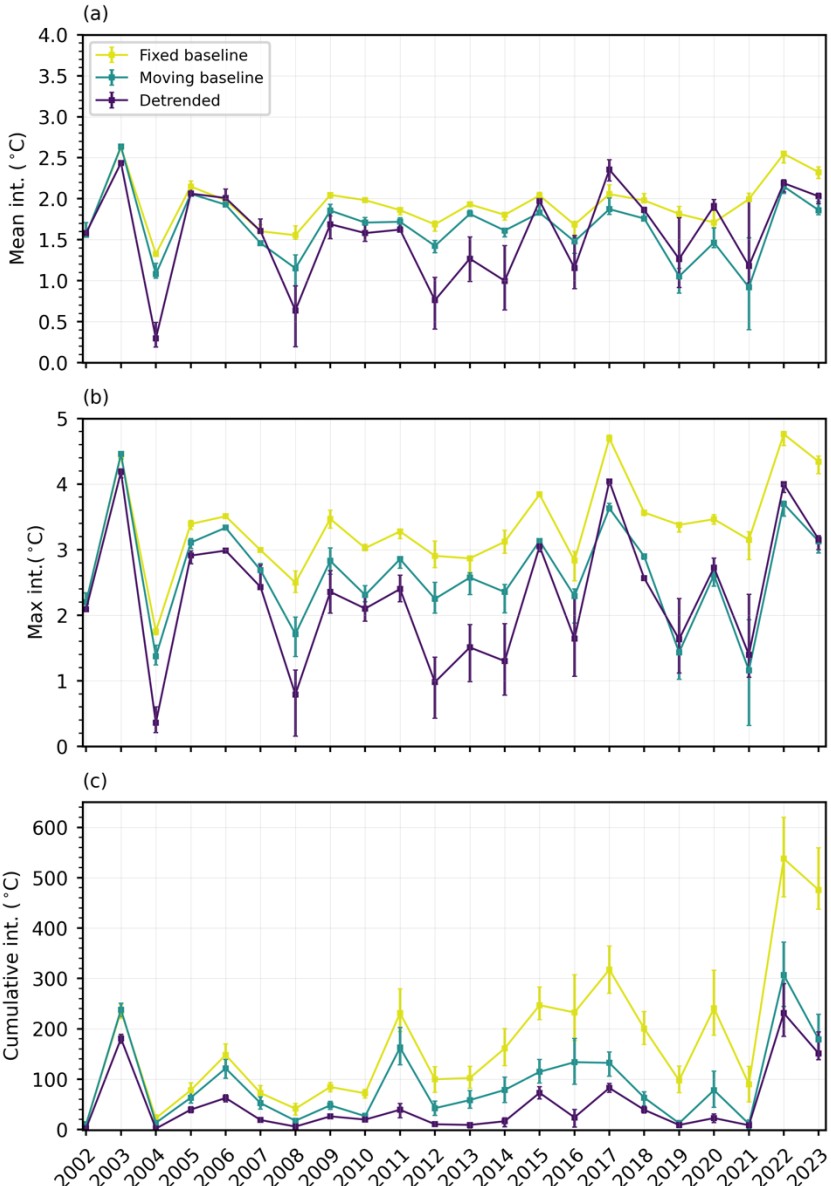

**Figure 4: Spatially averaged metrics of the MHWs detected in the Balearic Sea from 2002 to 2023 using three detection methods: fixed baseline, moving baseline and detrended data. (a) Annual average of the mean intensity of MHW events. (b) Annual average of the maximum intensity of MHW events. (c) Annual average of the cumulative intensity of MHW events.**

In terms of mean intensity, the detrended approach shows the greatest variability across years, while mean intensities calculated using the moving baseline are consistently lower than those obtained with a fixed baseline. Mean intensities in 2006, 2017, and 2020 are higher when computed with detrended data than with the other methods, whereas in 2007, 2015, and 2023, the mean intensity from the detrended approach exceeds only the moving baseline values.



For maximum intensity, the fixed baseline consistently provides the highest values across all years, while detrended intensities exceed those from the moving baseline only in 2017 and 2022. Cumulative intensity shows an increasing trend with the fixed baseline approach, which is less apparent with the other methods.

**4.3 Spatial variability of marine heatwaves**

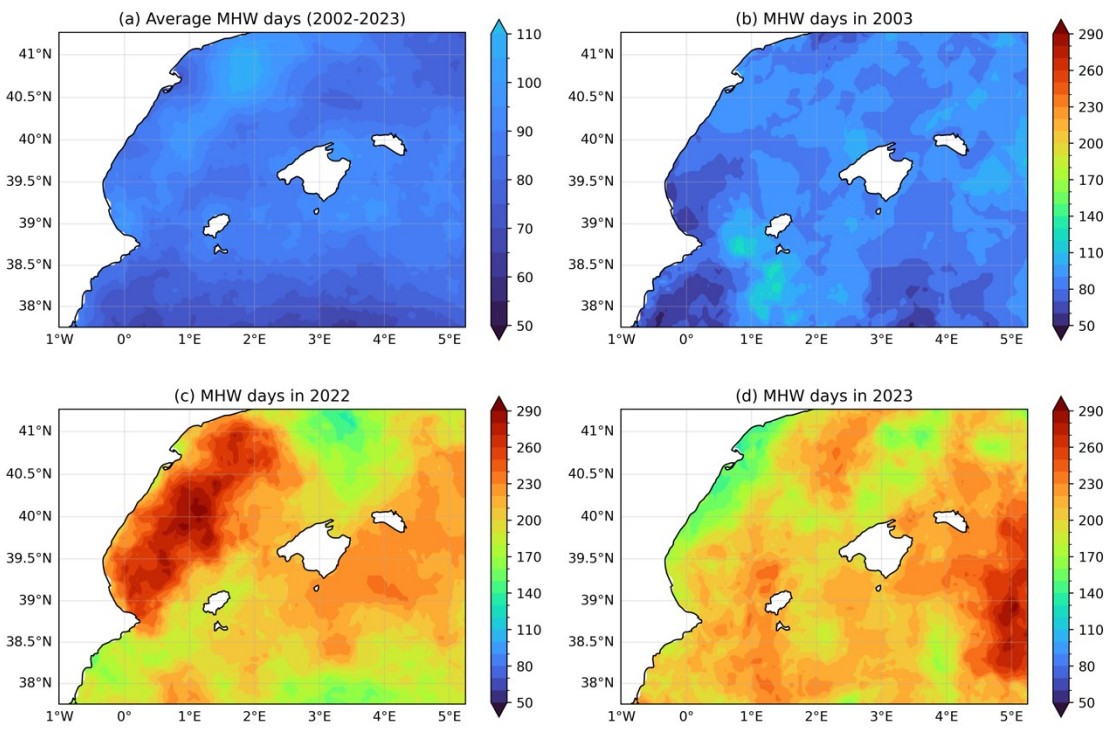

**Figure 5: Spatial distribution of the annual number of days with MHWs detected using the fixed baseline method for: (a) the temporal average from 2002 to 2023, (b) 2003, (c) 2022 and (d) 2023.**

The spatial distribution of the annual number of days with MHWs on average and for 2003, 2022 and 2023 is represented in
Figs. 5 (fixed baseline), 6 (moving baseline) and 7 (detrended data). The temporally averaged number of MHW days obtained with a fixed baseline (Fig. 5a) and with a moving baseline (Fig. 6a) shows slightly higher values around the islands and to the east of the study region. This is not detected by the detrended method (Fig. 7a), where the higher number of MHW days is located north of Mallorca, around 41ºN. With the fixed baseline, 10-30% of the days of the year undergo MHWs, this value is reduced with a moving baseline and detrended data to 9-16% and 3-8%, respectively (note the different colour scales used in
Figs. 5-7).

The 2003 MHWs detected with a fixed baseline were more frequent to the south and southwest of Ibiza, with maximum values of 140 days under MHWs (Fig. 5b). The 2022 MHWs were more frequent in the area along the Iberian Peninsula with a total of 290 days under MHW conditions (Fig. 5c). For this same baseline, the 2023 number of days under MHWs shows similar





maximum values to those in 2022 but, in this case, the area with the highest number of days with MHWs is in the eastern part
of the study region (Fig. 5d). Both 2022 and 2023 maximum values of MHW days double the highest values obtained in 2003
(Fig. 5b).

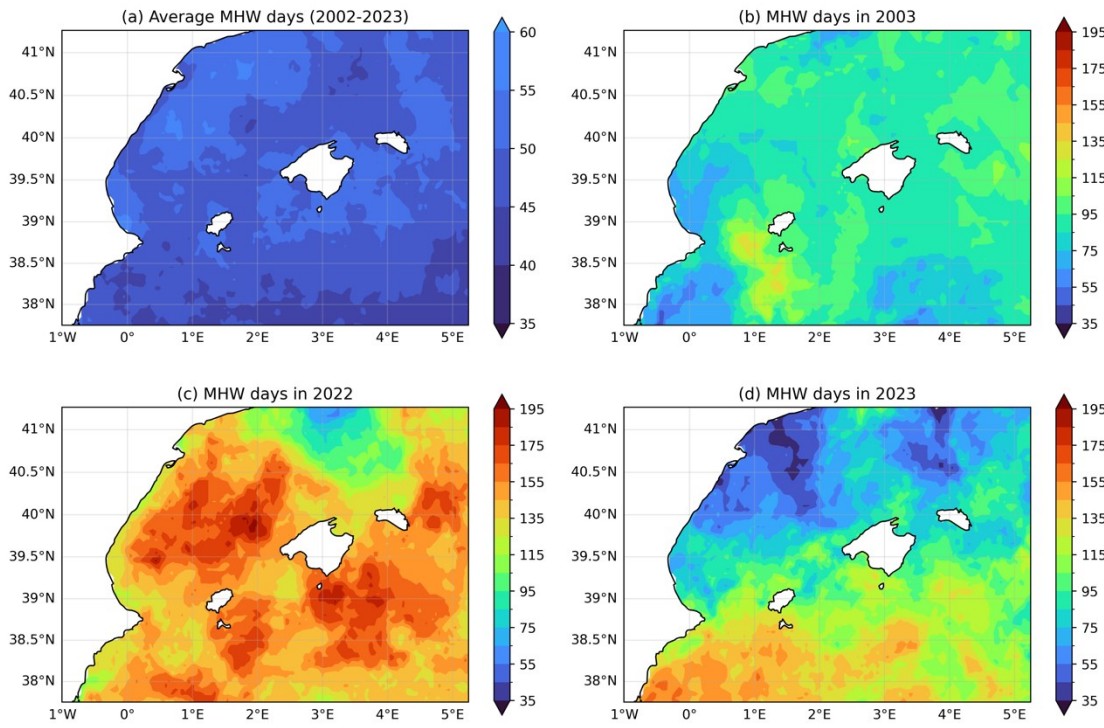

**Figure 6: Spatial distribution of the annual number of days with MHWs detected using the moving baseline method for: (a) the temporal average from 2002 to 2023, (b) 2003, (c) 2022 and (d) 2023.**

With a moving baseline, in 2022 the number of MHW days is more evenly distributed along the region, with minimum values
north of Mallorca (Fig. 6c). The years 2003 and 2023 show a similar range of values for the number of MHW days, but their
spatial distribution differs significantly. Whereas in 2003 the distribution is more homogenous, with most areas having between
a total of 80 and 100 MHW days, in 2023 we can find greater spatial variability, with a higher number of MHW days in the
south and lower in the north of the study region (Figs. 6b and 6d).

With the detrended method, 2003 had a similar number of MHW days in all the Balearic Sea, but with slightly higher values
around the islands and to the northeast of the region (Fig. 7b). In 2022, the area northwest of Mallorca shows the highest values
with areas reaching more than 155 days undergoing MHW conditions (Fig. 7c). The MHWs in 2023 exhibit slightly lower
values than in 2003, the areas with the highest MHW days are located around the island and to the south of the region but we
also find a patch north of Mallorca (Fig. 7d).





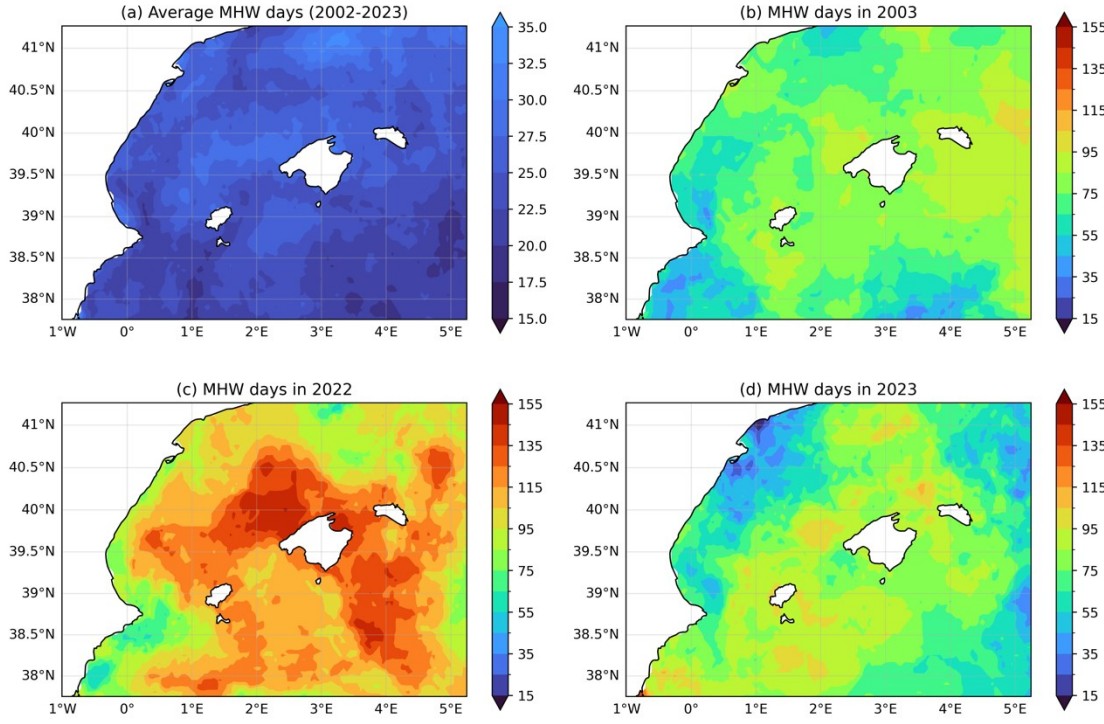


**Figure 7: Spatial distribution of the annual number of days with MHWs detected using the detrended method for: (a) the temporal average from 2002 to 2023, (b) 2003, (c) 2022 and (d) 2023.**

The spatial distribution of MHWs days in 2003 remains relatively unchanged with the three detection methods, in opposition to 2022 and 2023. In 2022, when using a moving baseline, MHW days appear in the form of evenly dispersed patches across

the region. When using the other two methods, localised areas of higher number of MHW days are present. With the fixed baseline this area is near the coast of the Iberian Peninsula, and with detrended data this area is northwest of Mallorca. In 2023, the easternmost part of the study region is identified as the area with the highest number of MHW days using the fixed baseline, while also displaying one of the lowest numbers of MHW days when using detrended data.

The areas where the warming trend is higher show on average more MHW days with a fixed baseline than we observe with

the other two methods (Figs. 5a, 6a and 7a). This is the case of the warming spot on the eastern part of the study region along the Iberian Peninsula coast.

## 4.4 Implications of different definitions

When comparing the MHWs detected using three different methods (i.e. fixed baseline (Hobday et al., 2016), moving baseline (Rosselló et al., 2023) and detrending (Martínez et al., 2023)) we find differences amongst them. With the fixed and moving

baselines, we obtain similar results for all the metrics in the first part of the studied period until 2009. This is expected since this period is when the baselines overlap the most. The difference becomes larger after 2009 when the fixed baseline yields



higher MHW days, duration and intensity than the other methods. The 2003 MHWs are widely considered one of the greatest extreme temperature events occurred in the Mediterranean in the last few decades (Holbrook et al., 2019), causing a mass mortality effect in benthic communities (Crisci et al., 2011; Marbà et al., 2015). Using the year 2003 to compare latter MHWs,

when a fixed baseline is employed the annual number of MHW days in 2003 is surpassed 8 times (in 2011, 2015, 2016, 2017, 2018, 2020, 2022 and 2023), but only 3 times when using a moving baseline (in 2011, 2022 and 2023) (Figs. 3 and 4). This discrepancy is a consequence of the impact of the warming trend on the fixed baseline method. Detrended data highlights 2017 as a year with short intense MHWs, 2023 as a year with long and frequent events of lower intensity, and 2003 and 2022 as years with both intense and long events. In general, this method shows lower frequency and duration than the other two

methods. Regarding intensity, there are changes, but depending on the year the mean and maximum intensity values are higher or lower than with the other methods. This is consistent with Oliver (2019), who found that the warming trends mostly affected the frequency and duration of the detected MHWs, while having a lesser impact on their intensity.

These findings highlight the importance of selecting a MHW detection method that aligns with the specific impacts intended for the study (Amaya et al., 2023; Darmaraki et al., 2024). In terms of ecosystems, this will depend on the ability of adaptation

of certain species or communities. A fixed baseline can highlight when temperatures exceed critical thresholds that historically have not been surpassed. For less adaptable species, which are more sensitive to specific temperature thresholds, a fixed baseline could better indicate when these critical points are reached, signalling potential risks to their survival. Moving baselines and detrended data adjust for the ongoing changes in the climate, and species with higher adaptability might show resilience to gradual shifts of the temperature but still be vulnerable when exposed to extreme temperature rises.

In the Mediterranean, Garrabou et al. (2022) found that mass mortality events related to MHWs that occurred from 2015 to 2019 affected a number of taxa comparable to those that happened in 2003, with Cnidaria and Bryozoa being the most affected groups. This is relevant because both 2015 and 2017 are highlighted as years with high intensity using the detrended method and with fixed baseline, whereas with a moving baseline the most prominent MHWs appear in 2016. Mortality is not the only consequence of MHWs, the function and behaviours of the organisms can be affected too. Marbà et al. (2015) reported that

while mortality increased in response to a quick rise in temperature, growth, survival, fertility, migration, and phenology were negatively altered in response to the warming trend. They also determined that some taxonomic groups (Ascidiacea, Crustacea, Echinodermata, fish, and phytoplankton) are more sensitive than others (Cnidaria, Mollusca and Porifera, and seagrasses) to changes in temperature. One extreme case of this are species that not only can adapt to warming oceans but even benefit from them (Boudouresque et al., 2024), provided a lethal temperature anomaly is not surpassed. These thermophilic species can, in

some cases, expand their distribution, displacing native species more sensitive to changes in temperatures (Wesselmann et al., 2024).

Expanding our knowledge about this topic can increase our effectiveness in prevention via prediction models and real-time monitoring, both of which enable a quick response in situations where the management of resources is needed. One case where this is particularly crucial is in fisheries. Extreme conditions like MHWs can cause reductions in recruitment and selecting an

adequate fishing strategy can ensure the safety of the stock (Caputi et al., 2016). There have been big efforts in developing





prediction models for MHWs in the last few years, however, most of them are still considered to be in an 'experimental phase' (Cornwall, 2023).

## 5. Conclusions

This study provides an evaluation of sea surface temperature trends and marine heatwaves in the Balearic Sea from 42 years
of SST satellite observations. We compare the results obtained by applying three different MHW detection methods (fixed baseline, moving baseline, and detrended data). The analysis reveals a significant warming trend of 0.036 ± 0.001 °C/year in the Balearic Sea, with the highest increases observed south of Menorca and northwest of Mallorca. The comparison of MHW detection methods highlights the impact of methodological choices on MHW characterisation. The fixed baseline method indicates a substantial increase in both the number and duration of MHWs over time, peaking in recent years. In contrast, the
moving baseline shows less pronounced increases in MHW parameters over time, suggesting that adjusting the baseline to recent years can mitigate the apparent increase in MHWs detected by the fixed baseline. The detrended data approach further isolates MHWs from underlying temperature trends, resulting in fewer detected MHW days and shorter event durations. With this method, the mean and maximum intensities of MHWs were highest in 2003, 2017, and 2022. The cumulative intensity, which considers both the duration and magnitude of MHWs, was particularly high in 2022, reflecting prolonged and intense
heatwave conditions. But regardless of the method employed, 2003 and 2022 stand out as the most prominent events in the Balearic Sea over the past couple of decades.

Overall, this study reveals the importance of method selection in MHW detection, demonstrating that the choice of baseline can significantly influence the annual number of days under MHWs, their duration and intensities. While a fixed baseline detects an increase in MHWs over time, a moving baseline and the detrending method provide alternative approaches that aim
to isolate extreme events from long-term trends.

## AUTHOR CONTRIBUTIONS

All authors conceptualized the study. B. F.-A. performed the data analysis and wrote the first draft. All authors contributed to the preparation of the final draft.

## COMPETING INTERESTS

The authors declare that they have no conflict of interest.



## FINANCIAL SUPPORT

This study is contribution number 14 to the project ObsSea4Clim "Ocean observations and indicators for climate and assessments" funded by the European Union (Grant Agreement number: 101136548). B. F.-A. received support through a JAE-Intro scholarship granted by the Spanish National Research Council (CSIC). B. B.-L. is funded by the Balearic
Government Vicenç Mut program (grant PD/008/2022). We also acknowledge support of the contract C3S2_520_CSIC/SC2 funded by European Copernicus Climate Change Service (C3S) implemented by European Centre for Medium-Range Weather Forecasts (ECMWF).

## ACKNOWLEDGEMENTS

The present research was carried out within the framework of the activities of the Spanish Government through the "María de
Maeztu Centre of Excellence" accreditation to IMEDEA (CSIC-UIB) (CEX2021- 001198). This work represents a contribution to CSIC Interdisciplinary Thematic Platform (PTI) Teledetección (PTI-TELEDETECT).

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
