# Peer review of "Tracking Marine Heatwaves in the Balearic Sea: Temperature Trends and the Role of Detection Methods"

_EGUsphere, 2024_

## Author Comment (AC1)

We would like to thank the reviewer for their efforts in revising our manuscript and their feedback. We appreciate the time and effort invested in reviewing our manuscript and are pleased that the reviewer finds the topic relevant and the methodology appropriate.

Below, we response comment by comment and outline the corresponding changes made to the manuscript. Reviewer comments are shown in black with the author response in blue. The lines refer to the clean manuscript to which the reviews have been applied.

**Reviewer Comments:**

**Reviewer 1**

In this work the authors analyse the impact of different methodologies, namely the reference baseline method, in the MHW detection and characteristics in the Balearic Sea. This is a topic under open discussion among the MHW research community. The authors conclude that the choice of the detection method is crucial depending on the research intended objectives, but that's not a novelty as some recent studies have already underlined. The manuscript is worth publishing but needs some improvement, outlined below.

**We thank the reviewer for the positive assessment of our work and the recognition of its relevance to the ongoing discussion in the MHW research community.**

The introduction section should more in detail discuss studies on Mediterranean MHWs, that have very recently received growing interest in scientific literature, as the manuscript refers to a Mediterranean subregion. Recent papers discuss the consideration of spatial requirements for MHWs, build MHW catalogues or MHW drivers.

**We agree with this suggestion. We have revised the introduction to mention some relevant recent literature (Darmaraki et al., 2024; Simon et al., 2023) to better contextualise the study**:

*"In their comprehensive review, Darmaraki et al. (2024) gather the findings about MHWs events and their drivers in the Mediterranean Sea, also commenting on known impacts across diverse marine organisms. The authors argue that despite many advances in the field, a single, uniform detection framework remains both technically challenging and, despite its benefits for study-comparability, potentially inappropriate, since it should be adapted to the intrinsic local conditions. MHWs in the Mediterranean Sea are mainly generated due to long periods of anomalously high atmospheric pressure that lead to weakened values of wind speed (Darmaraki et al., 2024; Simon et al., 2023)."* [Lines 66-72]

**And we have further discussed on the MHW drivers in the Mediterranean:**

*"Regionally, MHWs in the western Mediterranean basin are generally more intense and frequent, while in the eastern basin they are typically longer (Hamdeno and Alvera-Azcaráte, 2023; Simon et la., 2022). In the western basin, the events are usually driven by large positive heat-flux anomalies (enhanced short-wave radiation and reduced latent heat loss), whereas in the east, they often coincide with decreased upward long-wave radiation (Hamdeno and Alvera-Azcaráte, 2023; Simon et al., 2023)."* [Lines 74-78]

The authors use appropriate methodology to assess those impacts and properly discuss them. But the last part of section 4.4 (from line 285) looks on definition impacts in biodiversity more like as a review of the state-of-the-art of MHW biological impacts. These section needs to be, in part, rewritten to better highlight the authors results.

**We appreciate this comment and have revised Section 4.4 to better focus on how our methodological comparison impacts the interpretation of potential biological consequences. While some background on biological impacts is still provided for context, we have reduced the general review tone and emphasized how the differences in MHW detection approaches could lead to varying assessments of ecological risk. Now this part reads as follows:**

*"To illustrate the ecological relevance of these methodological choices, in the Mediterranean, Garrabou et al. (2022) reported that mass mortality events from 2015 to 2019 affected a similar number of taxa as in 2003, with Cnidaria and Bryozoa being the most affected groups. A similar impact was also detected in the study regions, where the intense summer MHW of 2017 led to a proliferation of mucilaginous algae, which triggered mass mortality of gorgonian species and affected sponge communities (Bensoussan et al., 2019). This is relevant because both 2015 and 2017 are highlighted as years with high intensity using the detrended method and with fixed baseline, whereas with a moving baseline the most prominent MHWs appear in 2016. Such effects had been previously observed following the 2003 MHW, when the same mucilaginous algae caused coral bleaching and necrosis in sponges and coralline algae in the Ligurian Sea, Northwest Mediterranean (Schiaparelli et al., 2007). During 2019, Hamdeno and Alvera-Azcaráte (2023) found that anomalously high temperatures coincided with a decrease of surface chlorophyll in the western Mediterranean. Interestingly, 2019 is not a year where we detect high MHW activity. Future work could compare chlorophyll concentration from 2019 with those from strong MHW years such as 2017 or 2022."* [Lines 294-304]

Minor comments

Line 40: Please correct "in studies in studies"

**The repetition has been corrected.** [Line 40]

Line 47: Please correct "underlaying"

**Corrected.** [Line 47]

Line 65: Change "air-sea heat fluxes and producing a decrease" to "air-sea heat fluxes and produce a decrease"

**The sentence has been revised as suggested.** [Line 72]

2.4 Marine heat waves metrics

The category list and table 1 are unnecessary as they are well described in the Hobday references. Please consider adding the necessary definitions in a new paragraph.

**Following the reviewer comment, we have removed the category list and table 1 and replaced with a concise paragraph summarizing the key metrics:**

*"For each MHW event, we compute its duration and its mean, maximum and cumulative intensity (Hobday et al., 2016). The mean intensity is defined as the average SST anomaly of a MHW event, the maximum intensity is the highest SST anomaly reached during a MHW episode, and the cumulative intensity is the sum of the daily SST anomalies during a MHW event. We also compute the total number of days per year with MHWs. [...] Following Hobday et al. (2018), MHWs are classified into four categories according to the extent to which they exceed the 90th percentile threshold: Moderate (1-2 times), Strong (2-3 times), Severe (3-4 times), and Extreme (more than 4 times)."* [Lines 122-125 and 127-128]

We hope these revisions adequately address the reviewer's concerns and improve the clarity and scientific contribution of our manuscript. We remain grateful for the helpful feedback.

Sincerely,
Blanca Fernández-Álvarez, on behalf of all co-authors

---

## Author Comment (AC2)

We would like to thank the reviewer devoting time to revise our manuscript and their feedback. We believe that addressing their feedback has significantly strengthened the overall quality of the paper.

Below, we provide a point-by-point reply to each of the reviewer's comments, along with an outline of the corresponding changes made to the manuscript. Reviewer comments are shown in black with the author response in blue. The lines refer to the clean manuscript to which the reviews have been applied.

**Reviewer 2**

This article focuses on the Balearic Sea, a critical sub-region of the western Mediterranean, highlighting its ecological importance and the increased risk of species loss due to climate change. The main objective of the study is to assess trends in sea surface temperatures (SST) and marine heatwaves (MHWs) obtained by different methods, using 42 years of SST satellite observations (1982-2023).

Although the study is well written and makes a valuable contribution to the field, it needs some major concerns to increase its clarity and overall impact. These improvements are described in detail below:

We thank the reviewer for the positive assessment of our work and for recognising its relevance and clarity. We appreciate their constructive suggestions, which have helped to improve the manuscript.

**Major comments:**

- The authors follow the definition of MHW by Hobday et al. (2016). In several places in the manuscript, it is mentioned that Hobday et al. (2016) proposed to calculate the MHW climatology based on SST from 1982 to 2001. This is not accurate, as they suggested using an SST for at least 30 years to derive climatological baselines. In this paper, they used the years 1982 to 2011 to calculate the MHWs of 2003, 2011 and 2012, which was quite appropriate. In addition, there are several studies that have followed Hobday et al. (2016) and used 30 or more years of climatology in the Mediterranean and different regional seas. They have avoided reaching saturation (year-round MHWs) by using the appropriate baseline for their study period and at the same time their work met the criteria suggested by Hobday et al. (2016). Some of these articles are ([1–7]).

This comment should definitely be taken into account as it changes the results, especially in the part of the comparison between the different methods, and could also mean changes in the conclusion.

**We agree with the reviewer, and we have corrected the manuscript accordingly. We now explicitly state that Hobday et al. (2016) recommend using at least 30 years of data to define the climatological baseline and that the 1982–2011 period is used in their examples.**

**This has been modified in methods:**

*"Following the definition proposed by Hobday et al. (2016), a MHW is described as a high-temperature event where a threshold value is exceeded for a minimum of five consecutive days. The threshold is usually set at the 90th percentile of the distribution at each grid point over a specific climatology constructed using at least 30 years of data. The fixed baseline refers to the use of a common climatology in the MHW identification process, computed here using data from the period between 1982 and 2011 (Hobday et al., 2016)."* [Lines 104-108]

**And also in the introduction:**

*"All the studies previously mentioned apply the standard approach to detect MHWs, which involves calculating SST anomalies over a fixed historical climatology constructed from a common baseline, often based on the 1982–2011 period (Hobday et al., 2016)."* [Lines 43-45]

**In accordance with this, we recalculated the fixed baseline and detrended MHWs using the 30-year (1982–2011) climatology and updated the results and figures. This correction led to slight changes on some MHW metrics (e.g. Tables S1 and S2) but had a minor effect on the overall conclusions. The results and discussion were modified where necessary to reflect this update. In addition, we added a comment in the introduction about the efforts done by other authors to avoid saturation:**

*"Reaching saturation is not inherent to the fixed ‑baseline method. By extending the baseline to 30 years or more (e.g., Ibrahim et al., 2021; Juza et al., 2022; Darmaraki et al., 2024), saturation can be avoided. However, as MHW characteristics remain influenced by long ‑term ocean warming, it induces an apparent upward trend in both duration and intensity (Mohamed et al., 2022; 2023)."* [Lines 47-50]

*"Reaching saturation is not inherent to the fixed baseline method. By extending the baseline to 30 years or more (e.g., Ibrahim et al., 2021; Juza et al., 2022; Darmaraki et al., 2024), saturation can be avoided. However, as MHW characteristics remain influenced by long ‑term ocean warming, it induces an apparent upward trend in both duration and intensity (Mohamed et al., 2022; 2023)."* [Lines 47 -50]

-  The introduction would benefit from a more detailed discussion of recent studies on MHWs in the Mediterranean region. As the MS focuses on a sub-region of the Mediterranean, it would be good to have a comparison with the other Mediterranean regions from the literature ([1,3,4]).

**We agree with the reviewer and have expanded the introduction to include a more comprehensive overview of MHW studies across different Mediterranean subregions. We now cite and compare findings from recent studies covering the western and eastern Mediterranean basins. The following paragraph was added:**

*"Regionally, MHWs in the western Mediterranean Basin are generally more intense and frequent, while in the eastern basin they are typically longer (Hamdeno and Alvera-Azcaráte, 2023; Simon et la., 2022). In the western basin, the events are usually driven by large positive heat-flux anomalies (enhanced short-wave radiation and reduced latent heat loss), whereas in the east, they often coincide with decreased upward long-wave radiation (Hamdeno and Alvera-Azcaráte, 2023; Simon et al., 2023)."* [Lines 74-78]

**Additional references suggested by the reviewer have been included where appropriate** (for instance: Hamdeno and Alvera-Azcaráte, 2023 [Lines 75, 78 and 302]; Ibrahim et al., 2021 [Line 48]; Mohamed et al., 2022 [Line 50]; Mohamed et al., 2023 [Line 50]).

**Minor notes:**

**Figures and tables:**

- Table 1: Consider removing this table or replacing it with a short paragraph in the text.

**We removed Table 1 and incorporated its contents into a concise paragraph summarizing the key metrics and categories:**

*"For each MHW event, we compute its duration and its mean, maximum and cumulative intensity (Hobday et al., 2016). The mean intensity is defined as the average SST anomaly of a MHW event, the maximum intensity is the highest SST anomaly reached during a MHW episode, and the cumulative intensity is the sum of the daily SST anomalies during a MHW event. We also compute the total number of days per year with MHWs. [...] Following Hobday et al. (2018), MHWs are classified into four categories according to the extent to which they exceed the 90th percentile threshold: Moderate (1-2 times), Strong (2-3 times), Severe (3-4 times), and Extreme (more than 4 times)."* [Lines 122-125 and 127-128]

- Figure 2, panel (b): I recommend removing this panel and keeping only panel (a and c). If the authors have a good reason to keep panel (b), I suggest replacing the term "residuals" of SST with "deseasonalized" SST, since the term "residuals" is typically associated with sea level data that include tidal and residual components.

**We retained panel (b) as it supports the description of the SST anomaly decomposition. However, we have replaced the term "residuals" with "deseasonalized SST" to ensure proper terminology and clarity.** [Line 130 and 133]

- Figure 3 and 4: Could the author add the grid on the figures and add the tick on the x-axis of panel (a) of the figures.

**We updated Figures 3 and 4 to include gridlines and improved the visibility of x-axis ticks on Figure 3a and Figure 4a and 4b.** [Lines 180 and 212]

- Figure 5: For clarity, use a different color scheme in panel (a) that better differentiates between high and low values.

- Figures 6 and 7: The same suggestions apply as for Figure 5.

**We updated panel (a) in Figures 5, 6 and 7 to better illustrate the variability for the average MHW days with each baseline.** [Lines 225, 241 and 254]

**Abstract:**

- In the abstract, please indicate the time period used to calculate the SST trends before mentioning the results of it.

**We updated the abstract, now it reads as follows:**

*"For the period between 1982 and 2023, our analysis reveals ..."* [Line 16]

**Data and methods:**

- Line 96: as previously mentioned Hobday et. al (2016) suggested using 30 for the climatology baseline calculations and for their study they used data from 1982 to 2011, please correct this here and everywhere else in the MS.

**This has been corrected and now reads:**

*"Following the definition proposed by Hobday et al. (2016), […] specific climatology constructed using at least 30 years of data. The fixed baseline refers to the use of a common climatology in the MHW identification process, computed here using data from the period between 1982 and 2011 (Hobday et al., 2016)."* [Lines 104-108]

**Results:**

- Figures 3 and 4: It is quite interesting that the year 2022 shows an increase in MHW characteristics for the three MHW detection methods, especially for the method that uses detrended data. Does this mean that the strong occurrence of MHW in 2022 was not primarily caused by global SST trends. Did the authors try to identify the reasons for this interesting result?

**This is indeed interesting. We argue that the strength of the use of multiple detection methods lies in their ability to distinguish between SST warming trend and extreme events. The fact that 2022 stands out even when using the detrended method indicates that the event was not a consequence of the SST warming trend but, rather, driven by exceptional atmospheric and oceanic conditions. The 2022 MHW has been previously documented (Marullo et al., 2023; Juza et al., 2024; McAdam et al., 2024; Pirro et al., 2024; Trigo et al., 2025) and several studies point to the persistence of anticyclonic atmospheric conditions over the western**

**Mediterranean, which supressed vertical mixing and anomalously high net shortwave radiation. We have added this observation to the manuscript:**

*"Notably, the 2022 MHW was consistently detected by the fixed baseline, moving baseline and detrending methods, underscoring its strength. The 2022 MHW has been previously documented (Marullo et al., 2023; Juza et la., 2024; Pirro et al., 2024; McAdam et al., 2024; Trigo et al., 2025) and was linked to an exceptional anticyclonic atmospheric blocking over the western Mediterranean, the anticyclonic conditions persisted during the winter months, suppressing wind-driven mixing during fall/winter (Marullo et al., 2023; Trigo et al., 2025)."* Lines [281–286]

- There is an error in the numbering of the headings. There is section 2 and then section 4, please correct this.

**Corrected**

**Suggested References:**

1. Hamdeno M., Alvera-Azcaráte A. // Front. Mar. Sci. 2023. v. 10. p. 1093760.https://www.frontiersin.org/articles/10.3389/fmars.2023.1093760/full. (Accessed June 10, 2023)
2. Hamdeno M., Alvera-Azcárate A., Krokos G., Hoteit I. Marine Heatwaves in the Red Sea and their Relationship to Different Climate Modes: A Case Study of the 2010 Events in the Northern Red Sea // Remote Sensing/Climate and modes of variability. 2024 (Accessed February 8, 2024).https://egusphere.copernicus.org/preprints/2024/egusphere-2024-355/. (Accessed February 8, 2024)
3. Ibrahim O., Mohamed B., Nagy H. // JMSE. 2021. V. 9. № 6. P. 643.https://www.mdpi.com/2077-1312/9/6/643. (Accessed June 14, 2021)
4. Galli G., Solidoro C., Lovato T. // Front. Mar. Sci. 2017. v. 4. p. 136.http://journal.frontiersin.org/article/10.3389/fmars.2017.00136/full. (Accessed June 15, 2021)
5. Mohamed B., Nilsen F., Skogseth R. // Frontiers in Marine Science. 2022. V. 9. https://www.frontiersin.org/article/10.3389/fmars.2022.821646. (Accessed June 12, 2022)
6. Mohamed B., Ibrahim O., Nagy H. // Remote Sensing. 2022. V. 14. № 10. P. 2383.https://www.mdpi.com/2072-4292/14/10/2383. (Accessed May 27, 2022)
7. Mohamed B., Barth A., Alvera-Azcárate A. // Front. Mar. Sci. 2023. V. 10. https://www.frontiersin.org/articles/10.3389/fmars.2023.1258117. (Accessed June 14, 2024)

**As mentioned above, most of these references are now included in the revised version of the manuscript.**

We hope these revisions adequately address the reviewer's concerns and improve the clarity and scientific contribution of our manuscript. We remain grateful for the helpful feedback.

Sincerely,
Blanca Fernández-Álvarez, on behalf of all co-authors